# Plasmonic ommatidia for lensless compound-eye vision

Leonard C. Kogos [1], Yunzhe Li [1], Jianing Liu[1], Yuyu Li [1], Lei Tian [1] & Roberto Paiella [1✉]

The vision system of arthropods such as insects and crustaceans is based on the compound-eye architecture, consisting of a dense array of individual imaging elements (ommatidia) pointing along different directions. This arrangement is particularly attractive for imaging applications requiring extreme size miniaturization, wide-angle fields of view, and high sensitivity to motion. However, the implementation of cameras directly mimicking the eyes of common arthropods is complicated by their curved geometry. Here, we describe a lensless planar architecture, where each pixel of a standard image-sensor array is coated with an ensemble of metallic plasmonic nanostructures that only transmits light incident along a small geometrically-tunable distribution of angles. A set of near-infrared devices providing directional photodetection peaked at different angles is designed, fabricated, and tested. Computational imaging techniques are then employed to demonstrate the ability of these devices to reconstruct high-quality images of relatively complex objects.

[1] Department of Electrical and Computer Engineering and Photonics Center, Boston University, 8 Saint Mary's Street, Boston, MA 02215, USA.
✉email: rpaiella@bu.edu

Traditional cameras used for common imaging applications consist of one or multiple lenses projecting an image of the object of interest onto an array of photodetectors. This configuration, similar to the human eye architecture, can provide excellent spatial resolution, but suffers from a fundamental tradeoff between small size and wide field-of-view, originating from aberration effects at large angles of incidence. In nature, the solution devised by evolution to address this issue is the compound eye[1], which in fact is universally found among the smallest animal species, such as insects and crustaceans (Fig. 1a). While different types exist, its basic architecture consists of an array of many imaging elements called ommatidia pointing along different directions (Fig. 1b), each collecting a single point of information about the scene being imaged. Typical ommatidia found in the apposition compound eye include a facet lens, a crystalline cone, a waveguiding fiber (rhabdom), and photoreceptor cells. These elements can be packed in extremely compact volumes providing nearly full hemispherical vision with no aberration. With this arrangement, all objects in the field-of-view are also automatically in focus at all times (i.e., the depth of field is essentially infinite), regardless of their distance from the camera and without the need for any focal-plane readjustment, leading to exceptional acuity to motion. As a result of these unique attributes, optoelectronic compound-eye cameras are ideally suited to address a wide range of imaging applications where extreme size miniaturization, wide-angle fields of view, and high temporal resolution are of particular importance. Specific examples include chip-on-the-tip endoscopy, concealed surveillance, wearable and swallowable cameras, and machine vision for obstacle avoidance and autonomous navigation, especially in drones.

These considerations have motivated significant research efforts on the development of novel cameras directly inspired by the compound-eye vision modality. Most prior implementations have been based on planar[2–4] or curved[4–11] arrays of microlenses combined with carefully aligned image-sensor arrays. The curved geometry directly mimics the compound-eye architecture of common arthropods, but is complicated by limited compatibility with standard microelectronic circuits, which are traditionally based on planar substrates. As a result, it requires either the introduction of bulky optical relay systems[5,8,9,11] or the development of complex fabrication and packaging processes to produce photodetector arrays and readout electronics on a curved surface[6,7,10]. A possible implementation of a flat compound-eye camera is shown in Fig. 1c, where the photodetector/microlens pair in each pixel detects light incident along a different direction determined by the position of the photodetector within the focal plane of the microlens[3]. However, the field-of-view in this geometry is severely constrained by optical crosstalk between neighboring pixels at large angles of incidence (as illustrated by the dashed line in Fig. 1c), which can lead to the formation of ghost images unless interpixel blocking layers are employed. Either way, even with multiple lens arrays the maximum achievable field-of-view is limited by the $f$-number of the microlenses to values below ±35°[4]. In a somewhat related technology, each photodetector is stacked with two diffraction gratings on top of one another to produce a sinusoidal dependence of detected signal on angle of incidence[12]. While also capable of significant miniaturization, this approach is not directly based on the compound-eye vision modality and its imaging capabilities depend on the results of a more complex global deconvolution procedure. Additionally, its maximum reported field-of-view (<±50°) is once again limited by design and fabrication constraints. More recently, the use of optical phased arrays for angle-sensitive photodetection has also been reported[13], which allows for dynamic tuning of the angle of peak detection but requires a local laser oscillator for heterodyne mixing. Finally, another related device from the recent literature is a photodetector consisting of two closely spaced nanowires that can directly measure the polar illumination angle, for applications such as triangulation and optical ranging (rather than full image reconstruction)[14].

Here, we describe a compound-eye camera based on a fundamentally different approach that can provide wide-angles field-of-view (over ±75°) on a flat substrate. Its key innovation is the integration of each pixel of a standard image-sensor array with a specially designed metasurface (ensemble of subwavelength optical nanostructures) that only allows for the detection of light incident along a small, geometrically tunable distribution of angles, whereas light incident along all other directions is reflected (Fig. 1d). Computational imaging techniques are then employed to enable image reconstruction from the combined signals of the individual sensors. With this approach, ultrathin planar cameras can be developed without any lenses, featuring all

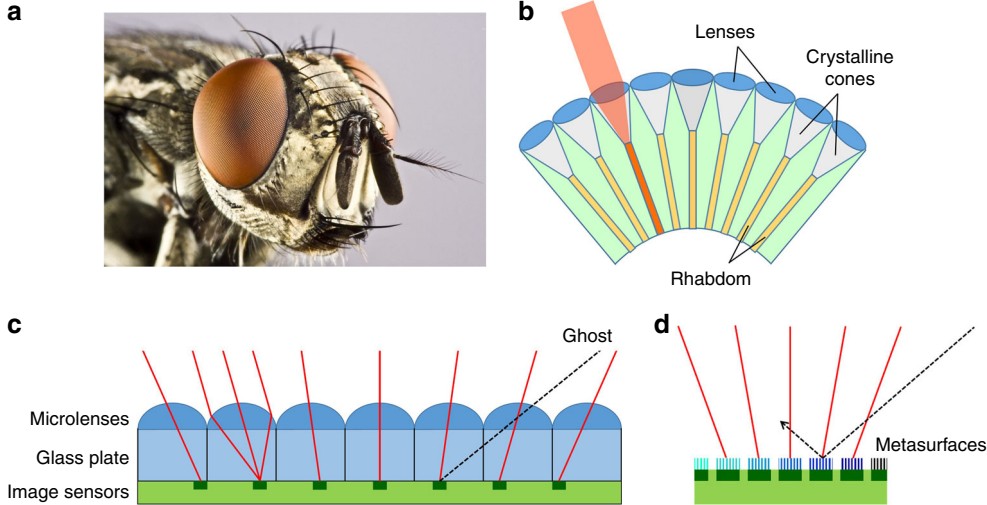

**Fig. 1 Compound eyes. a** Micrograph of the compound eyes of a horse fly. Used with permission (copyright: Michael Biehler/123RF.COM). **b** Schematic illustration of the apposition compound-eye architecture. **c** Artificial compound-eye camera based on a planar microlens array and a photodetector array separated by a glass plate. By design the two arrays have different periodicities, so that each sensor detects light incident along a different direction. **d** Compound-eye camera based on the angle-sensitive metasurfaces developed in the present work, where only light incident along a different direction is transmitted into each image sensor.

the aforementioned desirable attributes of compound eyes and providing further miniaturization compared to previous implementations. In particular, the lack of a microlens array reduces both the camera thickness and the required spacing between neighboring photodetectors, allowing for higher density and therefore higher resolution. The metasurfaces can be fabricated directly on existing CMOS/CCD image-sensor arrays using standard lithographic techniques, with straightforward alignment to their respective pixels and full suppression of interpixel crosstalk. In the following, we present the design, fabrication, and characterization of a representative set of infrared devices providing directional photodetection peaked at different angles, based on metallic plasmonic nanostructures combined with simple Ge photoconductors. A computational imaging framework is then developed to investigate the imaging capabilities of cameras consisting of complete arrays of such devices. These simulations are carried out using both calculated and measured angular response patterns of the experimental devices, together with their interpolations for all other pixels in the array. The key conclusion is that high-quality images of relatively complex objects can be reconstructed over a wide field-of-view of ±75°, with realistic operational characteristics including number of pixels, signal-to-noise ratio, and illumination bandwidth.

## Results

**Metasurface design**. The principle of operation of the angle-sensitive devices developed in this work is illustrated in Fig. 2. The photodetector active material (a Ge photoconductor) is coated with a composite metasurface consisting of a metal film stacked with an array of rectangular metallic nanoparticles (NPs). The metasurface comprises three different sections—a periodic grating coupler, a grating reflector, and a set of slits through the underlying metal film. Gold is used as the choice material for all metallic features, due to its favorable plasmonic properties at infrared wavelengths[15]. Two dielectric layers (SiO$_2$) are also introduced immediately below and above the Au film, to provide electrical insulation from the active layer and to control the film-NP coupling, respectively. Because the metal film is optically thick (100 nm), photodetection can only take place through an indirect process where light incident at the desired angle is first diffracted by the NPs (in the periodic grating coupler section) into surface plasmon polaritons (SPPs)—i.e., guided electromagnetic waves propagating along the Au-air interface. A small number of sub-wavelength slits in the metal film are then used to scatter these SPPs into radiation propagating predominantly into the absorbing active layer. As a result, a photocurrent signal is produced between two biased electrodes located across the metasurface.

The incident angle of peak detection is controlled by the grating coupler period $\Lambda$. Specifically, SPPs propagating along the $\mp x$ direction of Fig. 2a, b can be excited via first-order diffraction of light incident (on the $x$–$z$ plane) at the equal and opposite angles $\pm\theta_p$ determined by the diffraction condition $(2\pi\sin\theta_p)/\lambda_0 - 2\pi/\Lambda = -2\pi/\lambda_{SPP}$, where $\lambda_0$ and $\lambda_{SPP}$ are the wavelengths of the incident light and excited SPPs, respectively. Light incident at any other angle is instead completely reflected or diffracted away from the surface (in particular, the excitation of SPPs by all higher orders of diffraction is avoided by keeping $\Lambda$ smaller than $\lambda_{SPP}$).

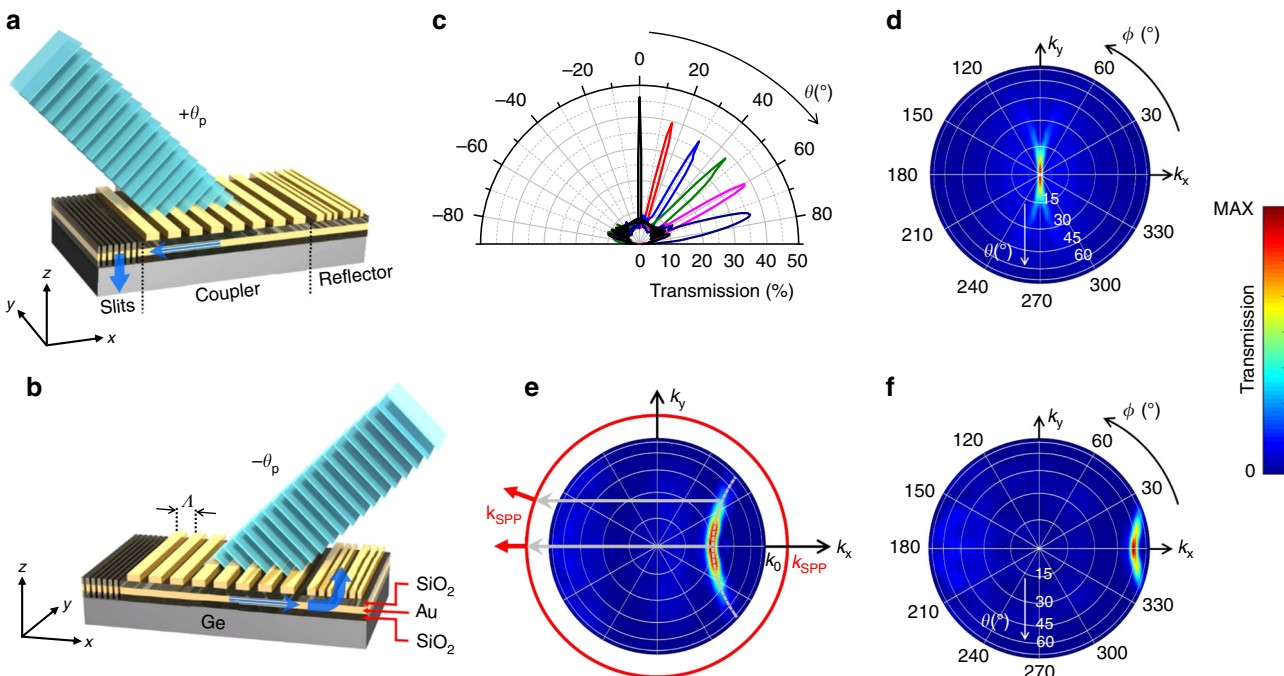

**Fig. 2 Angle-sensitive metasurfaces. a**, **b** Schematic illustrations of the metasurface geometry and principle of operation. Light incident at the desired detection angle $+\theta_p$ (**a**) is diffracted by the grating coupler into SPPs propagating towards the slits, where they are preferentially scattered into the absorbing substrate. Light incident at the opposite angle $-\theta_p$ (**b**) is diffracted by the NP array into SPPs propagating toward the grating reflector, where they are diffracted back into radiation. Light incident at any other angle is instead completely reflected or diffracted away from the surface. **c** Calculated optical transmission coefficient at $\lambda_0 = 1550$ nm through six different metasurfaces for $p$ polarized light versus angle of incidence $\theta$ on the $x$–$z$ plane. The grating coupler period (number of NPs) ranges from 1465 to 745 nm (15 to 29) in order of increasing angle of peak detection. The NP widths vary between 250 and 570 nm. **d**–**f** Transmission coefficient of three metasurfaces from **c** as a function of both polar $\theta$ and azimuthal $\phi$ illumination angles, summed over $xz$ and $yz$ polarizations. In each map, $k_x$ and $k_y$ are the in-plane components of the incident-light wavevector, and the color scale is normalized to the maximum (MAX) transmission value. In **e**, the solid red circle of radius $k_{SPP}$ indicates the available SPP modes on the top metal surface; the dashed curved line highlights the incident directions of peak transmission; the horizontal grey arrows (having length $2\pi/\Lambda$) illustrate how light incident along these directions can excite SPPs by negative-first-order diffraction; and the red arrows show the directions of propagation of the excited SPPs.

The selective detection of only one incident direction (e.g., $+\theta_p$) is then obtained by surrounding the grating coupler with the slits on one side (in the $-x$ direction) and the grating reflector on the other side (in the $+x$ direction). The reflector is another array of rectangular NPs designed to scatter the incoming SPPs into light radiating away from the sample near the surface normal direction. With this arrangement, the SPPs excited by incident light at $+\theta_p$ propagate toward the slits, where they are preferentially scattered into the substrate and produce a photocurrent (Fig. 2a and Supplementary Movie 1). The SPPs excited by incident light at $-\theta_p$ propagate towards the grating reflector, where they are diffracted back into free space (Fig. 2b and Supplementary Movie 2). As a result, the metasurface-coated photodetectors are functionally equivalent to the ommatidia of the apposition compound eye, while maintaining the planar geometry of standard image sensor arrays.

The metasurfaces just described rely on a number of key ideas from plasmonics and nanophotonics, here applied to a novel device functionality (directional filtering). First, the ability of subwavelength slits to efficiently couple SPPs to radiation is well established in the context of extraordinary optical transmission[16] and has already been exploited for various applications[17–19]. In detail, when an SPP propagating on the top metal surface reaches the slit boundaries, a line of in-plane oscillating dipoles is effectively produced across the slit, which will then emit radiation mostly propagating into the higher-index substrate. The same behavior in reverse has also been employed for the efficient excitation of SPPs on the top surface of a perforated metal film, via illumination from the back side[20–22]. Second, the design of the grating reflector is based on the notion of metasurfaces with a linear phase gradient[23,24], where composite asymmetric unit cells are used to suppress all orders of diffraction $q$ except for $q = -1$ (see Supplementary Note 2 and Supplementary Fig. 2). As a result, SPP transmission (which is equivalent to zero-order diffraction) is effectively forbidden in this NP array, so that the incident SPPs from the grating coupler (as in Fig. 2b) can be completely scattered into radiation with the smallest possible number of periods. In a photodetector array, any SPP transmitted across the reflector of one pixel may be scattered and detected into a neighboring pixel. The use of a linear phase gradient is therefore favorable to avoid spurious photocurrent signals produced by light incident at $-\theta_p$ (see Supplementary Fig. 3). Similarly, if the $q = +1$ order were allowed, near-normal incident light could be partially diffracted by the grating reflector into SPPs also propagating directly into a neighboring pixel, where again they could produce an undesired signal (in contrast, any SPP excited in the grating reflector by $q = -1$ diffraction will propagate along the $-x$ direction across the entire NP array, where it can experience near complete attenuation through absorption and scattering before reaching the slits on the other side).

Several devices based on the geometry just described, each providing peak photodetection at a different angle $\theta_p$, have been designed using full-wave electromagnetic simulations based on the finite difference time domain (FDTD) method. In addition to the grating coupler period $\Lambda$, key design parameters include the number of NPs (which can be optimized for maximum peak transmission) and the NP width (which should be selected to maximize the grating diffraction efficiency, while at the same time avoiding any significant coupling between SPPs and localized plasmonic resonances supported by the NPs); more details can be found in Supplementary Note 1 and Supplementary Fig. 1. Figure 2c shows the calculated $p$-polarized power transmission coefficient for a set of optimized metasurfaces at $\lambda_0 = 1550$ nm, as a function of polar angle of incidence $\theta$ on the $x$–$z$ plane (the relevant geometrical parameters are listed in Supplementary

Note 3 and Supplementary Table 1). If the metasurfaces are fabricated on a photodetector active material, the detected signal is proportional to their transmission coefficient. The devices of Fig. 2c can therefore provide tunable directional photodetection, with a wide tuning range for the angle of peak detection $\theta_p$ of $\pm 75°$ and narrow angular resolution, ranging from 3° to 14° full-width-at-half-maximum (FWHM) as $\theta_p$ is increased. The peak transmission coefficient $T_p$ is in the range of 35–45% for all designs considered, with a peak-to-average-background ratio of about 6. In passing, it should be noted that in the structure with $\theta_p = 0°$, the grating coupler is surrounded by slits on both sides (since the desired angular response is symmetric), leading to a somewhat larger value of $T_p$. For $s$-polarized light, the transmission through the same metasurfaces is isotropic and significant smaller, <0.2% at all angles (see Supplementary Fig. 4 and discussion below).

The full angular response patterns of the same devices are shown in the color maps of Fig. 2d–f and Supplementary Fig. 5, where the metasurface transmission coefficients (computed with a reciprocity-based method and summed over both polarizations) are plotted as a function of both polar $\theta$ and azimuthal $\phi$ illumination angles. In each map, the directions of high transmission form a C-shaped region within the full hemisphere, which is indicative of first-order diffraction of the incident light into SPPs of different wavevectors $\mathbf{k_{SPP}}$. Specifically, the C shape is determined by the reciprocal-space distribution of the available SPP modes at $\lambda_0$ (red circle in Fig. 2e), translated by the lattice vector $\hat{x} 2\pi/\Lambda$ of the grating coupler (as shown by the horizontal arrows in the same figure). This behavior clearly increases the range of incident directions detected by each pixel. Importantly, however, the computational imaging techniques described below allow reconstructing images with higher resolution compared to the single-pixel angular selectivity, if devices with appropriate overlaps in their angular responses are combined.

For any incident direction, the metasurface transmission for $xz$-polarized light (i.e., with electric field on the $x$–$z$ plane) is again much larger than for $yz$-polarized light (see Supplementary Note 4). This behavior originates from the polarization properties of SPPs. In general, SPPs possess an in-plane electric-field component that is parallel to their direction of propagation[15]. Therefore, in the geometry under study, $xz$-polarized incident light is most effective at exciting SPPs propagating at a small angle with respect to the $x$ axis, and vice versa. In the same geometry, where the slits are linear and oriented along the $y$ direction, only SPPs with a large $x$ (i.e., perpendicular) component of the electric field can be efficiently coupled into radiation through the aforementioned excitation of oscillating dipoles across the slits[22]. It follows from these considerations that the SPP modes that are more strongly scattered by the slits into the absorbing substrate are also more effectively excited by $xz$-polarized (as compared to $yz$-polarized) incident light. The same considerations also explain why the metasurface transmission within the C-shaped regions of Fig. 2d–f decreases with increasing azimuthal angle $\phi$ of the incident light: the larger $\phi$, the smaller the $x$-components of the wavevector $\mathbf{k_{SPP}}$ and electric field of the correspondingly excited SPPs (see red arrows in Fig. 2e). The intrinsic polarization dependence of the devices of Fig. 2 limits their overall sensitivity for typical imaging applications involving unpolarized light. At the same time, it could be exploited in conjunction with computational imaging techniques to enable polarization vision, which offers several desirable features such as reduced glare and improved contrast[25]. Alternatively, polarization independent angle-sensitive photodetectors could also be designed with more complex metasurfaces, e.g., using two-dimensional NP arrays that allow for independent phase and polarization control[26,27].

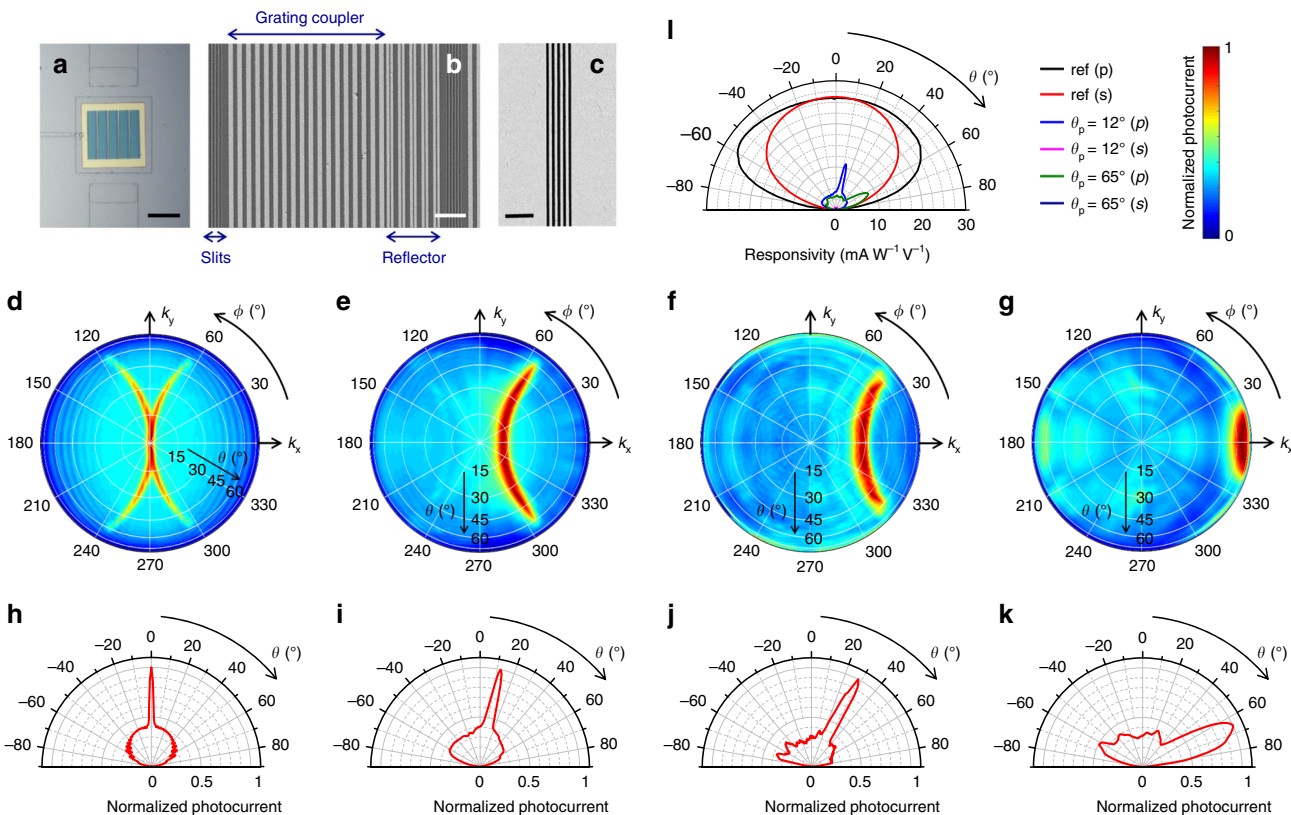

**Fig. 3 Measurement results. a–c** Optical (**a**) and SEM (**b, c**) images of representative experimental samples. The scale bar is 100 μm in **a**, 4 μm in **b**, and 2 μm in **c**. In **a** the entire metasurface of a complete device is seen through a Ti window covering the entire sample, which is introduced to avoid spurious photocurrent signals. The image of **c** was taken before fabrication of the NP array. **d–g** Measured angular dependence of the photocurrent of four devices based on the structures of Fig. 2, providing peak response near $\theta_p \approx 0°$ (**d**), 12° (**e**), 28° (**f**), and 65° (**g**). In each plot, the photocurrent is normalized to the peak value. SEM images reveal some deviations in the array periods and NP widths from their target design values. The measured values are $\Lambda = 1440$, 1180, 1030, and 775 nm and $w = 240$, 560, 526, and 256 nm for the devices of panels **d, e, f**, and **g**, respectively. **h–k** Line scans along the $\phi = 0°$ direction from the maps of **d–g**, respectively. **l** p- (i.e., xz-) and s- (i.e., yz-) polarized responsivity versus polar angle of incidence on the x–z plane, measured with three different samples: a reference device without any metal film and NP array, and two metasurface-coated devices providing peak detection at $\theta_p = 12°$ and 65°, respectively. Source data for panels **d–g** are provided as Source Data files.

**Experimental results.** The metasurfaces of Fig. 2 can be applied to any planar photodetector technology regardless of its operation principles. Here we use metal-semiconductor-metal (MSM) Ge photoconductors, where a photocurrent signal is collected across two biased electrodes deposited on the top surface of a Ge substrate. The angle-sensitive metasurface is patterned on the active region between the two metal contacts. While photodiodes generally offer higher performance, MSM photodetectors are particularly simple to fabricate and therefore provide a convenient platform to investigate the metasurface development. To simplify the angle-resolved photocurrent measurements, we also use relatively large active areas: in each device, the separation between the two electrodes is $d \approx 300$ μm, and the metasurface consists of a few (5–6) identical repetitions of a same structure based on the design of Fig. 2a, with the grating reflector of one section immediately adjacent to the slits of the next section. Representative optical and scanning electron microscopy (SEM) images are presented in Fig. 3, showing a complete device (Fig. 3a), a metasurface section (Fig. 3b), and a set of slits (Fig. 3c).

Angle-resolved photocurrent measurements with these devices show highly directional response in good agreement with the simulations (Fig. 3d–k and Supplementary Fig. 8). In these measurements, each device is illuminated with laser light at 1550 nm wavelength, and the polar and azimuthal angles of incidence are varied, respectively, by rotating the focusing optics about the sample and by rotating the sample about its surface

normal. Two orthogonally polarized angular-response maps are recorded for each sample, and their sums are plotted in Fig. 3d–g. Consistent with the discussion above, the measurement results indicate that the strongest photocurrent signal is obtained when the incident light is xz-polarized, whereas the yz-polarized contribution is essentially negligible (see Supplementary Fig. 7). Each map of Fig. 3 features the expected C-shaped region of high responsivity, centered near the designed polar angle of maximum metasurface transmission $\theta_p$ (0, 12, 28, and 65° for panels d, e, f, and g, respectively). The polar-angle selectivity (FWHM) of the same devices, measured from the $\phi = 0°$ horizontal line cut of each map shown in Fig. 3h–k, is in the range of 4–21° in order of increasing $\theta_p$. The peak-to-average-background ratio is ~3 for all devices. These measured characteristics are reasonably close to the calculated values from the simulation results of Fig. 2. The observed differences are mostly due to the presence of some surface roughness in the experimental samples (which can scatter some of the incident light into SPPs regardless of its direction of propagation), as well as small deviations in the array periods and NP widths (mostly affecting $\theta_p$). In any case, as described below, these experimental values are already fully adequate for high-quality image reconstruction.

To evaluate the peak transmission of the metasurfaces, otherwise identical bare samples without any metal film and NP array between the two electrodes were also fabricated and tested. Figure 3l shows the polar-angle-resolved

*p*- and *s*-polarized responsivity of one such sample, together with data measured with two metasurface devices. At their angles of peak detection of 12° and 65°, the *p*-polarized responsivities of the latter devices are reduced to ~42% and 36%, respectively, of the corresponding value from the bare sample, in excellent agreement with the simulation results of Fig. 2c. Unfortunately, a more extensive quantitative comparison among all experimental devices of Fig. 3 is not possible due to large variations in their dark resistances. Such variations were observed even among different samples based on the same design (including different bare samples), with the responsivity consistently increasing with dark resistance, and are possibly caused by fabrication-induced defects affecting the carrier density or promoting current leakage. As a result, in Fig. 3l we only include data measured with devices featuring the same dark resistance (~1.5 kΩ). It should also be noted that the values of peak responsivity per applied voltage shown in Fig. 3l (~10 mA $W^{-1}$ $V^{-1}$) are reasonable for this type of photodetectors, especially given their large inter-electrode separation $d \approx 300$ μm, which limits the photoconductive gain (proportional to $1/d^2$)[28].

**Image reconstruction**. Next we investigate the imaging capabilities of the angle-sensitive photodetectors just described. We consider a lensless compound-eye camera architecture consisting of a planar array of these devices, with each pixel providing directional photodetection peaked at a different combination of polar and azimuthal angles ($\theta_p$ and $\phi_p$, respectively). The value of $\theta_p$ can be controlled by varying the grating coupler design, as discussed above. For a fixed design, $\phi_p$ can be varied by simply rotating the entire metasurface about its surface normal on the corresponding photodetector. Using this pixel arrangement, we have conducted a series of numerical simulations by the following image-formation model. We consider objects sufficiently far away from the pixel array so that each angle corresponds uniquely to a different spatial point on the object (Fig. 4a). Each pixel integrates the total intensity detected according to its angular response.

The image-formation process can then be described by a linear matrix equation $\mathbf{y} = \mathbf{A}\mathbf{x}$, which relates the object's intensity distribution ($\mathbf{x}$) to the captured data ($\mathbf{y}$) by a sensing matrix ($\mathbf{A}$) (Fig. 4b). The angular response of each pixel forms a different row vector of $\mathbf{A}$, which quantifies the intensity contributions to the pixel signal from different points on the object[29]. To obtain the object's intensity distribution, we perform image reconstruction based on the truncated singular value decomposition (TSVD) technique[30]. The estimated object is given by $\hat{\mathbf{x}} = \sum_{l=1}^{L} \frac{1}{\sigma_l}(\mathbf{y}, \mathbf{u}_l)\mathbf{v}_l$, where $\mathbf{u}_l$ and $\mathbf{v}_l$ denote the *l*th left and right singular vector, respectively, and $\sigma_l$ is the corresponding singular value. *L* is a regularization parameter defining the number of singular vectors used in the TSVD solution, which is optimized by manual tuning based on visual inspection of the reconstructed image.

With this approach, we have validated the ability of both our designed and fabricated metasurfaces to enable complex image reconstruction. For the designed structures, the sensing matrix $\mathbf{A}$ is constructed from the calculated angular response maps of Fig. 2d–f and Supplementary Fig. 5, together with their interpolations for additional metasurfaces providing peaked transmission at different polar angles. The method for interpolating new pixel responses is detailed in Supplementary Note 6, and several interpolated examples are shown in Supplementary Figs. 9 and 10. The required number of different pixels is determined by calculating the superposition of all the pixel responses to ensure uniform field-of-view coverage, and through additional imaging simulations (see Supplementary Note 7 and Supplementary Fig. 11). Based on this analysis, we select $\Delta\theta_p = 1.5°$ and $\Delta\phi_p = 3°$ for the angular spacings between the directions of peak detection of consecutive pixels, which provide good image reconstruction quality with a reasonably small number of pixels (6240) covering the full ±75° field-of-view of the designed metasurfaces. With larger spacing in $\theta_p$, the reconstruction results suffer from radially oriented fringe artifacts due to missing coverage in the angular responses. With larger spacing in $\phi_p$, the resolution degrades especially in the high-polar-angle regions.

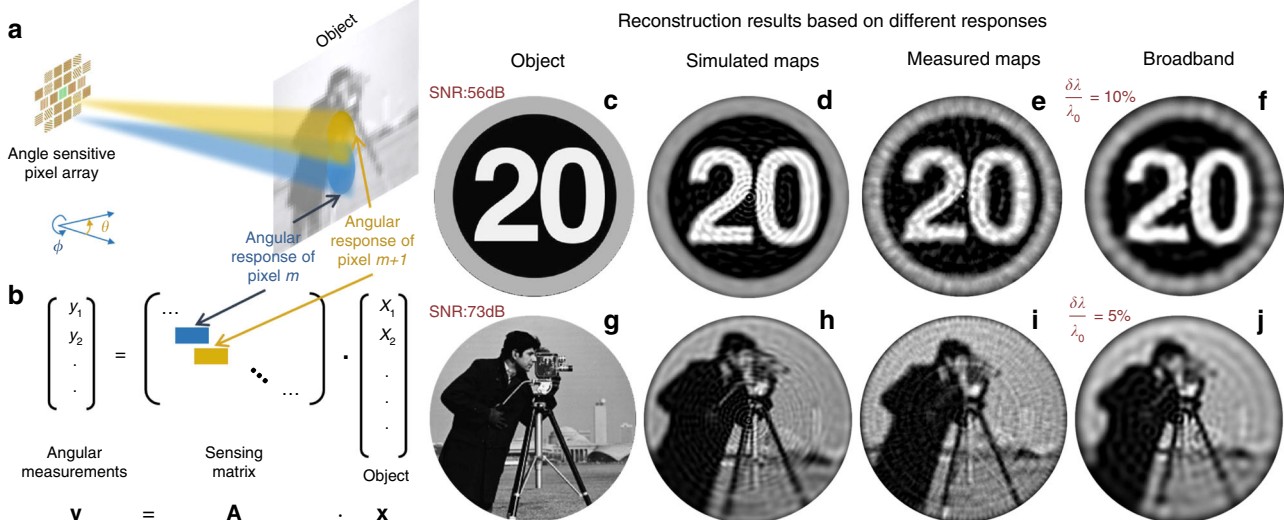

**Fig. 4 Data acquisition and image reconstruction. a** Schematic illustration of the imaging geometry. Each pixel integrates the incident light intensity from different directions according to its angular response. **b** Image-formation model. The pixel-array measurement is related to the object by a linear equation $\mathbf{y} = \mathbf{A}\mathbf{x}$, where the sensing matrix $\mathbf{A}$ contains the angular responses of all pixels. **c–f** Representative object (**c**) and corresponding image reconstruction results at SNR = 56 dB (**d–f**). **g–j** Example of a more complex object (**g**) and corresponding image reconstruction results at SNR = 73 dB (**h–j**). The original cameraman image (**g**) is used with permission from its copyright owner (Massachusetts Institute of Technology). The images of **d**, **h** are based on the simulated responsivity patterns of Fig. 2 with a 6240-pixel array at $\lambda_0 = 1550$ nm. The images of **e** and **i** are based on the experimental responsivity patterns of Fig. 3 with a 5280-pixel array at $\lambda_0 = 1550$ nm. The images of **f** and **j** are based on the simulated patterns under broadband illumination with bandwidth $\delta\lambda/\lambda_0 = 10$ % (**f**) and 5 % (**j**). The image reconstruction algorithm is made publicly available [https://doi.org/10.5281/zenodo.3634939].

A similar procedure with the same angular spacings is used to model the experimental devices, based on the measured angular response maps of Fig. 3d–g and Supplementary Fig. 8. The field-of-view for these interpolations is reduced to ±65° (limited by the maximum polar angle of peak detection measured with the present samples), spanned by 5280 pixels.

White Gaussian noise is also added to the captured data (the vectors **y**) to account for realistic photodetector performance (see Supplementary Note 8). In general, the signal-to-noise ratio (SNR) of a CCD/CMOS camera is limited by the saturation charge (full well capacity) of the individual pixels. Furthermore, it can be increased (by a factor of $\sqrt{N}$) by averaging the signals of ($N$) identical pixels, at the expense of a commensurate decrease in resolution and/or increase in active area. Here we use a baseline single-pixel SNR of 56 dB (i.e., $\mathbf{y}_{signal}/\mathbf{y}_{noise} = 631$), as reported in the literature with standard CMOS technology and optimized circuit designs, even for a pixel pitch as small as ~8 μm[31,32]. Additionally, we also conduct simulations for SNR = 63 and 73 dB, which can be achieved with larger arrays where each metasurface design is applied, respectively, to $N = 5$ and 50 pixels, whose signals are then binned together and averaged. The total number of pixels correspondingly increases up to about 260,000 and 310,000 (for cameras based on the measured and simulated devices, respectively, at the highest SNR of 73 dB), which is still well within the range of current CMOS technology. In passing, it should be noted that the same SNR values could also be achieved with several other combinations of pixel number, pixel dimensions, full well capacity, and bin size $N$.

The simulated imaging capabilities of our devices are illustrated in Fig. 4. Figure 4c–f contains results obtained for a relatively simple object (the speed-limit traffic sign of Fig. 4c), imaged at the baseline SNR of 56 dB. A more complex object (the cameraman picture of Fig. 4g) is considered in Fig. 4g–j, imaged at the larger SNR of 73 dB. Simulation results for arrays derived from both calculated (Fig. 4d, h) and measured (Fig. 4e, i) angular response maps are presented. High-quality image reconstruction is obtained in all cases, with the key features of both objects faithfully reproduced. Comparison between the results obtained with the calculated versus measured angular responses shows some loss of resolution in the latter case, caused by the lower angular selectivity and higher background levels of the experimental maps. In any case, these data clearly demonstrate the ability to reconstruct well-recognizable images even based on the measured characteristics of the fabricated devices. These observations are confirmed by extensive simulations carried out with several other objects of varying complexity at different SNRs, as shown in Supplementary Fig. 12.

Finally, we investigate how the imaging capabilities of the same devices are affected by the optical bandwidth $\delta\lambda$ of the incident light under polychromatic illumination. All the angular response maps employed so far are either computed or measured at a single wavelength – the target design value $\lambda_0 = 1550$ nm. At the same time, because of the diffractive nature of our metasurfaces, their transmission properties can be expected to vary with incident wavelength. Importantly, however, these variations can be accounted for in our computational imaging approach, so that well-recognizable images can also be reconstructed under reasonably polychromatic illumination with only a relatively small loss in resolution. In particular, if the incident spectrum extends over a finite bandwidth $\delta\lambda$, the main effect on the angular response of each device is a proportional broadening $\delta\theta_P$ of the detection peak. Using the diffraction condition above, we find $\delta\theta_P = \delta\lambda/\lambda_0(n_{SPP} + \sin\theta_P)/\cos\theta_P$, where $\theta_P$ is the polar angle of peak detection at $\lambda_0$, and the SPP effective index $n_{SPP} = \lambda_0/\lambda_{SPP}$ is ~1.06 in the metasurface designs of Fig. 2. Such broadening can

be included in the image-reconstruction simulations through a 2D convolution between the monochromatic pixel response and a Gaussian blurring kernel of width $\delta\theta_P$. Examples of images obtained with this approach applied to the simulated maps are shown in Fig. 4f, j, including the simple speed-limit sign imaged with a bandwidth $\delta\lambda/\lambda_0$ of 10% at 56-dB SNR (Fig. 4f) and the more complex cameraman picture for $\delta\lambda/\lambda_0 = 5\%$ and 73-dB SNR (Fig. 4j). The key features of both objects are once again well reproduced in the images. Additional examples can be found in Supplementary Fig. 13. The imaging situations considered in these simulations can be realized in practice by covering the entire camera array with a band-pass filter of 155- or 77-nm bandwidth. Larger operation bandwidths with higher image quality could be achieved using more complex gradient metasurfaces, with constituent elements designed to provide the same response at multiple wavelengths as in recent work towards broadband metalenses[33]. At the same time, it may also be possible to extract some information about the object's color distribution by first characterizing the spectral responses of each pixel followed by a multi-channel image reconstruction procedure, similar to recent work on diffractive-optics-based color imaging[34].

## Discussion

We have developed a new family of photodetectors that can mimic the behavior of the apposition-compound-eye ommatidia in a planar lensless format. Our numerical and experimental results show that these devices are promising for the development of a novel camera technology featuring the unique attributes of the compound eye, including small size, wide field-of-view, and high temporal bandwidth resulting from infinite depth of field. The maximum field-of-view considered in the present work (±75°) is determined by the largest polar angle of peak detection of the simulated devices, and can be further extended towards the full hemisphere with additional design optimizations. The ultimate limit is provided by the $\cos\theta$ decrease in intercepted optical power with increasing $\theta$ (e.g., by a factor of 10 for $\theta = 84°$), which can be partly compensated by using a higher readout gain and/or larger active area for the respective high-$\theta_P$ pixels. Similarly large fields of view can be achieved with optoelectronic apposition-compound-eye cameras based on curved arrays of image sensors and microlenses (e.g., ±80° in ref. [7]), at the expense however of significant fabrication complexity which also limits the number of pixels (e.g., 16 × 16 in the same work) and therefore the angular resolution. On the other hand, prior planar implementations are compatible with megapixel arrays, but suffer from limited fields of view (e.g., ±35° as discussed above[4]).

Our current devices are based on metallic NP arrays on near-infrared Ge MSM photoconductors, but can be developed with other materials platforms and photodetector technologies, including dielectric metasurfaces on visible-range Si photodiodes. The use of dielectric nanostructures is especially attractive as a way to minimize transmission losses, as well as allow for large-scale manufacturing in a CMOS-compatible foundry. The key operating principle of these devices (i.e., the integration of a photosensitive active layer with a metasurface that can selectively control the properties of the detected light) is also quite general, and provides extensive design degrees of freedom for improved performance or additional functionalities. For example, in the context of compound-eye vision, more complex NP arrays can be designed to enable directional photodetection with polarization independent transmission and broader operation bandwidth. Similar device architectures can also be envisioned for alternative applications, such as polarization sensing and direct wavefront reconstruction. Additional improvements in imaging quality and

capabilities are also possible from the computational side, by incorporating more advanced priors about the underlying objects via iterative reconstruction algorithms[35,36] or machine learning approaches[37,38]. Finally, this work highlights the tremendous opportunities offered by the synergistic combination of optical metasurface technology and computational imaging (as previously discussed in the microwave regime[39]). Metasurfaces allow controlling the flow (and in the present case the photodetection) of light in novel ways, and computational imaging techniques then provide a uniquely suited mechanism to extract maximal information from the resulting device outputs.

## Methods

**Design simulations**. The angle-sensitive metasurfaces are designed via FDTD simulations with a commercial software package (FDTD Solutions by Lumerical). Initial simulations consider a two-dimensional structure (on the plane perpendicular to the grating lines), consisting of the NP array and the $SiO_2/Au/SiO_2$ stack on a Ge substrate with the slits on one side and the grating reflector on the other. The metasurface is illuminated from above with a p- or s-polarized plane wave, and the transmission coefficient through a monitor plane parallel to the metal film in the Ge substrate is computed as a function of angle of incidence $\theta$. Periodic boundary conditions are applied to the lateral boundaries of the simulation region, whereas perfectly matched layers (PMLs) are used in the top and bottom surfaces. All relevant materials (Ge, $SiO_2$, Au) are described with a built-in database for their complex permittivity from the FDTD software. This procedure has been used to optimize the design parameters of all metasurfaces presented in this work, based on their transmittance at the desired detection angle $\theta_p$ and peak-to-background ratio. The full angular response pattern of each optimized device (as a function of both polar and azimuthal illumination angles) is then calculated with a reciprocity-based method, because computing the metasurface transmission in a three-dimensional structure for a sufficiently large number of incident directions would be prohibitively time consuming. In these simulations, an electric dipole source at $\lambda_0$ is positioned on the monitor plane in the Ge substrate (below the slits), and its far-field radiation pattern in the air above is computed (summed over different orthogonal orientations of the dipole). By reciprocity[40], this pattern is proportional to the local field intensity produced at the dipole location by an incident plane wave as a function of illumination angles. The line cut along the horizontal axis of each calculated radiation pattern is found to be in good agreement with the results of the 2D simulations just described (transmission versus angle of incidence on the x–z plane), which confirms the validity of this reciprocity-based approach. The proportionality factor between the radiation pattern and the transmission coefficient is then computed by comparing the 3D and 2D simulation results at $\theta_p$.

**Device fabrication**. The experimental samples are fabricated on nominally undoped (100)-oriented Ge substrates. A 60-nm-thick insulating $SiO_2$ layer is first deposited by rf sputtering, with two windows opened by photolithography and wet etching to allow for the subsequent formation of metallic contacts on the Ge surface. Next, the metasurface Au film and the device electrodes are introduced, and the slits are defined using a process based on electron-beam lithography (EBL) with a double resist layer of poly-methyl-methacrylate (PMMA) and hydrogen-silsesquioxane (HSQ)[41]. In this process: (1) lines at the desired slit positions are patterned by EBL in the HSQ layer (an image reversal electron-beam resist); (2) the exposed PMMA is removed by reactive ion etching (RIE); (3) a 100-nm-thick Au film with a 5-nm Ti adhesion layer is deposited by electron-beam evaporation; (4) an acetone bath is used to dissolve the residual PMMA and lift off the overlaying metals, leading to the formation of well-defined slits in the Ti/Au layer; (5) the same metallic layer is patterned by photolithography and wet etching to define the metasurface film and the device electrodes. Next, another 60-nm-thick $SiO_2$ layer is deposited on the entire substrate, with two windows opened over the electrodes to allow for wire bonding. The grating coupler and grating reflector (with each NP consisting of 5 nm of Ti and 50 nm of Au) are then fabricated by EBL with a single positive resist (PMMA), followed by electron-beam evaporation and liftoff. Finally, a Ti window with an opening over the entire metasurface is deposited on the top $SiO_2$ layer and patterned by photolithography. The function of this window is to suppress any spurious photocurrent signal that may otherwise be caused by light absorbed near the electrodes outside the Au film/NP-array stack. The completed devices are then mounted on a copper carrier and wire-bonded to two Au-coated ceramic plates.

**Device characterization**. The angle-resolved photocurrent measurements are carried out with a custom-built optical goniometer setup (see Supplementary Note 5 and Supplementary Fig. 6), where the device under study is illuminated with a diode laser at 1550-nm wavelength. The device is biased with a dc voltage while the incident light intensity is modulated at 1 kHz, so that the photocurrent can be measured separately from the dark current using a bias tee and lock-in amplifier. Typical values of the bias voltage and input optical power used in the measurements are 2 V and 0.5 mW, respectively. The laser output light is delivered with a polarization-maintaining fiber, which is mounted in a cage system together with a

2× beam expander (consisting of two lenses with 5- and 10-mm focal length), a half-wave plate to control the incident polarization, and an adjustable iris. The iris is used to ensure that the resulting focusing of the incident light on the photodetector is sufficiently small so as not to introduce any significant additional broadening in the measured angular response peaks. In order to vary the polar angle of incidence $\theta$, the entire cage is rotated with a piezo-controlled stage about the focal point of the output lens, where the device metasurface is carefully aligned using a microscope connected to a USB camera. The device is also mounted on another rotational stage that allows varying the azimuthal illumination angle $\phi$. For each setting of the latter stage, the half-wave plate in the cage system is rotated so that (at normal incidence) the input light is linearly polarized in the direction parallel or perpendicular to the grating lines. The polar angle is varied in steps of 1° between ±85° (the accessible limits of the goniometer setup), whereas the measured azimuthal angles range from 0° to 90° in steps of 5°. The remaining two quadrants of the angular response maps are then filled up based on the mirror symmetry of the device geometry under study with respect to the x–z plane. A linear interpolation procedure is also used in these color maps to include additional data points between the measured values of $\phi$ in steps of 1° (an example of a map without the interpolated points is shown in Supplementary Fig. 7). This procedure produces a smoother response map, as required for the image reconstruction simulations described above.

## Data availability
The datasets generated during the current study are available from the corresponding author upon reasonable request. The source data underlying Fig. 3d–g and Supplementary Fig. 8a–b are provided as Source Data files.

## Code availability
The custom codes in Matlab used for the image reconstruction tasks are available at https://doi.org/10.5281/zenodo.3634939.

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

## Acknowledgements

This work was supported by the National Science Foundation under Grant ECCS-1711156. The FDTD simulations were performed using the Shared Computing Cluster facility at Boston University.

## Author contributions

R.P. and L.T. directed the device development and computational imaging work, respectively. L.C.K., J.L., and Yuyu L. performed the device design, fabrication, and testing. Yunzhe L. carried out the imaging simulations. All authors were involved in the data analysis and paper preparation.

## Competing interests

The authors declare no competing interests.
