## [Peer Review File · Nature Communications]

Reviewers' comments:

Reviewer #1 (Remarks to the Author):

I enjoyed reading the paper entitled "Plasmonic ommatidia for lensless compound-eye vision" by Leonard Kogos and co-workers. The manuscript describes the design, fabrication, and characterization of an array of directional photodetectors. Good angle directionality is achieved with the help of a nanoscale metallic grating that couples light at a certain target frequency and incident angle to a planar plasmonic waveguide. This light is then coupled with a slit array into an underlying photodetector. A large array of the detector pixels is created where each pixel measures the signal originating from a certain incident angle. The work nicely brings together active areas of computational imaging and nanophotonics. The paper is well-written and worth publishing in Nature Comm after taking care of some suggested revisions.

Figure 1: It would suggest adding the Figure panels 2a and b to figure 1. It would be nice to illustrate show how the essential component of a compound eye (the directional coupler + detector) is now achieved with a plasmonic element right up front in the paper. This would leave some space in figure 2 for more quantitative data and simulations.

Figure 2: It would be helpful to have a more detailed, scientific drawing with the critical elements labeled in terms of materials and dimensions. An SEM image is provided later in the paper, but I initially had a hard time seeing exactly how the coupler is constructed and it was hard to check some of the basic numbers involved in the coupling. Ideally, the figure and methods section together would make it easy to reproduce the result. It would also be helpful to show a full-field simulation of the device (or grating coupler and slits separately) in the main text. I think this will help the read understand the coupling and relevant coupling efficiencies better. It would be great if a reasonable estimate of the measured responsivity could be provided based on the efficiencies of the coupler, slits, and detector.

As a minor note, I am not sure whether I would call the structure a metasurface. I think a plasmonic grating coupler best captures the function of the coupling device.

Reviewer #2 (Remarks to the Author):

The authors present in this work a very nice application of Plasmonics principles to achieve a bio-inspired device in imaging applications. From my point of view, this manuscript provides a very interesting example of what elementary Plasmonics studies can bring to real applications. In addition, the kind of device reported could be extremely useful in areas where the key issue is to access to wide-angles field of view.

The authors do all the way round from first principles, designing the gratings, slits... to the final device, including the image reconstruction process. The correspondence between experiments and theory is very good. This is not trivial, but a consequence of the theoretical analysis which through a rigorous optimization feeds the experiments.

One minor comment, it would be worth to include a table where a comparison between the performance of their device with others (or at least a single representative case) will be shown.

For all the above, I recommend this manuscript to be published on Nature Communications.

Plasmonic ommatidia for lensless compound-eye vision (Manuscript NCOMMS-19-24205)
– Response Letter

The reviewers' comments are listed below (in italics) together with our response and description of the corresponding changes made to the manuscript.

REVIEWER 1:

COMMENT: I enjoyed reading the paper entitled “Plasmonic ommatidia for lensless compound-eye vision” by Leonard Kogos and co-workers. The manuscript describes the design, fabrication, and characterization of an array of directional photodetectors. Good angle directionality is achieved with the help of a nanoscale metallic grating that couples light at a certain target frequency and incident angle to a planar plasmonic waveguide. This light is then coupled with a slit array into an underlying photodetector. A large array of the detector pixels is created where each pixel measures the signal originating from a certain incident angle. The work nicely brings together active areas of computational imaging and nanophotonics. The paper is well-written and worth publishing in Nature Comm after taking care of some suggested revisions.

RESPONSE: We thank the reviewer for the positive assessment of our work.

COMMENT: Figure 1: It would suggest adding the Figure panels 2a and b to figure 1. It would be nice to illustrate show how the essential component of a compound eye (the directional coupler + detector) is now achieved with a plasmonic element right up front in the paper. This would leave some space in figure 2 for more quantitative data and simulations.

RESPONSE: We agree with the reviewer that the introductory section of the paper would benefit from a plot that illustrates, right up front, how the ommatidia functionality is achieved in our work. However, panels (a) and (b) of Fig. 2 are designed to explain the detailed operation of our devices, which is described in a later section of the manuscript. Therefore, we have added instead a new schematic plot [panel (d) of revised Fig. 1], which shows how a flat lensless compound-eye camera can be implemented through the combination of plasmonic directional couplers and photodetectors, without going into the details of the coupler design.

COMMENT: Figure 2: It would be helpful to have a more detailed, scientific drawing with the critical elements labeled in terms of materials and dimensions. An SEM image is provided later in the paper, but I initially had a hard time seeing exactly how the coupler is constructed and it was hard to check some of the basic numbers involved in the coupling. Ideally, the figure and methods section together would make it easy to reproduce the result. It would also be helpful to show a full-field simulation of the device (or grating coupler and slits separately) in the main text. I think this will help the read understand the coupling and relevant coupling efficiencies better. It would be great if a reasonable estimate of the measured responsivity could be provided based on the efficiencies of the coupler, slits, and detector.

RESPONSE: We have modified panels (a) and (b) of Fig. 2 to include demarcation lines that clearly identify the different sections of our devices (grating coupler, grating reflector, and slits), and to label the constituent materials (Ge, SiO₂, Au) and key geometrical parameter (the grating coupler period Λ).

We also agree with the reviewer that presenting full-field simulation maps could be helpful. In fact, a particularly effective illustration of how our devices work can be obtained with a movie showing the calculated optical-field distribution around the metasurface as a function of time. Therefore, we have included two such movies as additional supplementary files. The first one shows how a pulse of light incident at the target detection angle is eventually transmitted across the metasurface (through the excitation of surface plasmon polaritons and their scattering by the slits into the underlying medium). The second movie shows how a pulse of light incident at the equal and opposite angle is instead reflected away from the device. Explicit reference to these movies has been added to the revised manuscript on pages 6 and 7.

The responsivity of our devices is determined by the intrinsic responsivity of the underlying photodetector, multiplied by the efficiencies of the grating coupler and slits. The former parameter is unrelated to the metasurfaces developed in the present work, which are compatible with essentially any planar photodetector technology. For the specific photodetectors used in our experiments (Ge MSM photoconductors), intrinsic responsivities on the order of a few 10 mA/W/V are measured, as shown in Fig. 3(I). The combined efficiency of the grating coupler and slits is in the range of 35–45 % depending on the specific metasurface design, according to both the simulation results of Fig. 2(c) and the experimental results of Fig. 3(I). These numbers are presented and discussed in the relevant paragraphs of the text, on pages 9, 13 and 14. It should also be noted that the efficiency of the slits is affected by the presence of the surrounding grating nanoparticles, which can reflect any surface plasmon transmitted across the slits back to the slits. As a result, in discussing efficiencies it is appropriate to treat the slits and coupler as a single combined system.

COMMENT: *As a minor note, I am not sure whether I would call the structure a metasurface. I think a plasmonic grating coupler best capture the function of the coupling device.*

RESPONSE: While our structures are different from typical metasurfaces reported in the literature, we believe that they still fit the general definition, as they consist of planar nanoparticle arrays of highly subwavelength thickness designed to manipulate the flow of light in novel and useful ways (specifically, by only transmitting light incident along a desired set of directions). We also note that our structures do indeed contain a “plasmonic grating coupler”, but only in a finite portion of their areas, combined with a phase gradient array on one side and a set of slits on the other. With this in mind, we have modified the initial description of the metasurface design (on page 5 of the revised manuscript) to read as follows: “**The photodetector active material (a Ge photoconductor) is coated with a composite metasurface consisting of a metal film stacked with an array of rectangular metallic nanoparticles (NPs). The metasurface comprises three different sections – a periodic grating coupler, a grating reflector, and a set of slits through the underlying metal film.**”

REVIEWER 2:

COMMENT: *The authors present in this work a very nice application of Plasmonics principles to achieve a bio-inspired device in imaging applications. From my point of view, this manuscript provides of a very interesting example of what elementary Plasmonics studies can bring to real applications. In addition, the kind of device reported could extremely useful in areas where the key issue is to access to wide-angles field of view.*

The authors do all the way round from first principles, designing the gratings, slits... to the final device, including the image reconstruction process. The correspondence between experiments and theory is very good. This is not trivial, but a consequence of the theoretical analysis which through a rigorous optimization feeds the experiments.

RESPONSE: We thank the reviewer for the positive assessment of our work.

COMMENT: *One minor comment, it would be worth to include a table where a comparison between the performance of their device with others (or at least a single representative case) will be shown.*

RESPONSE: Previous approaches towards optoelectronic compound-eye cameras differ widely from one another in terms of technology readiness level, degree to which they have been demonstrated, and details of what has been measured and/or modeled. Therefore, compiling a table that could provide a fair and comprehensive comparison would be problematic. Instead, we have added the following sentences in the Discussion section (on page 19 of the revised manuscript), where we discuss quantitatively key performance metrics of our devices in relation to representative prior works: “**Similarly large fields of view can be achieved with optoelectronic apposition-compound-eye cameras based on curved arrays of image sensors and microlenses (e.g., $\pm 80^\circ$ in ref. 7), at the expense however of significant fabrication complexity which also limits the number of pixels (e.g., 16×16 in the same work) and therefore the angular resolution. Vice versa, prior planar implementations are compatible with megapixel arrays, but suffer from limited fields of view (e.g., $\pm 35^\circ$ as discussed above [4]).**”

REVIEWERS' COMMENTS:

Reviewer #1 (Remarks to the Author):

I have read the revised manuscript and the response to the reviewer's comments in detail and I think everything looks very good and I believe the manuscript now deserves publication in Nature Communications.

Reviewer #2 (Remarks to the Author):

Like in my previous report, I recommend this manuscript to be published on Nature Communications.

The authors present a very nice application of Plasmonics principles to achieve a bio-inspired device in imaging applications. A very interesting example of what elementary Plasmonics studies can bring to real applications is provided. The device reported could be extremely useful in areas where the key issue is to access to wide-angles field of view.

The authors do all the way round from first principles, designing the gratings... to the final device, including the image reconstruction process. The correspondence between experiments and theory is very good. This is not trivial, but a consequence of the theoretical analysis which through a rigorous optimization feeds the experiments.

Sergio G Rodrigo

Plasmonic ommatidia for lensless compound-eye vision (Manuscript NCOMMS-19-24205A) – Response Letter

The reviewers' comments are listed below (in italics) together with our response.

REVIEWER 1:

COMMENT: I have read the revised manuscript and the response to the reviewer's comments in detail and I think everything looks very good and I believe the manuscript now deserves publication in Nature Communications.

RESPONSE: We thank the reviewer for the positive recommendation.

REVIEWER 2:

COMMENT: Like in my previous report, I recommend this manuscript to be published on Nature Communications.

The authors present a very nice application of Plasmonics principles to achieve a bio-inspired device in imaging applications. A very interesting example of what elementary Plasmonics studies can bring to real applications is provided. The device reported could be extremely useful in areas where the key issue is to access to wide-angles field of view.

The authors do all the way round from first principles, designing the gratings... to the final device, including the image reconstruction process. The correspondence between experiments and theory is very good. This is not trivial, but a consequence of the theoretical analysis which through a rigorous optimization feeds the experiments.

RESPONSE: We thank the reviewer for the positive recommendation.